

# Methodological considerations in assessment of language lateralisation with fMRI: a systematic review

Abigail R. Bradshaw, Dorothy V.M. Bishop and Zoe V.J. Woodhead

Department of Experimental Psychology, University of Oxford, Oxford, United Kingdom

## ABSTRACT

The involvement of the right and left hemispheres in mediating language functions has been measured in a variety of ways over the centuries since the relative dominance of the left hemisphere was first known. Functional magnetic resonance imaging (fMRI) presents a useful non-invasive method of assessing lateralisation that is being increasingly used in clinical practice and research. However, the methods used in the fMRI laterality literature currently are highly variable, making systematic comparisons across studies difficult. Here we consider the different methods of quantifying and classifying laterality that have been used in fMRI studies since 2000, with the aim of determining which give the most robust and reliable measurement. Recommendations are made with a view to informing future research to increase standardisation in fMRI laterality protocols. In particular, the findings reinforce the importance of threshold-independent methods for calculating laterality indices, and the benefits of assessing heterogeneity of language laterality across multiple regions of interest and tasks. This systematic review was registered as a protocol on Open Science Framework: https://osf.io/hyvc4/.

## INTRODUCTION

A wealth of evidence has demonstrated that language is predominantly mediated by the left cerebral hemisphere in the majority of individuals, a phenomenon known as hemispheric specialisation. This has been recently defined by *Tzourio-Mazoyer & Seghier (2016)* as "*the hosting by a given hemisphere of specialized networks that have specific functional properties and interact interhemispherically in a way that optimizes brain processing*". However, our understanding of the nature and correlates of such lateralisation has been hampered by the high level of variability in methods of its measurement, making integration of findings from across different studies difficult.

Non-invasive techniques for assessment of language lateralisation make it possible to probe the characteristics of language lateralisation in neurologically intact populations. Functional magnetic resonance imaging (fMRI) is a prominent non-invasive method that has been used to assess laterality. A laterality index ($LI$) is calculated based on a comparison

Corresponding author
Abigail R. Bradshaw,
abigail.bradshaw@psy.ox.ac.uk

of activation measures from each hemisphere, according to the following formula:

$$LI = \frac{L - R}{L + R}.$$

This calculates laterality as the difference between activity in each hemisphere ($L$ and $R$) divided by the total activity across the hemispheres. The $LI$ gives a single value indicating the relative strength of left and right hemisphere activation for an individual. $LI$ measurement may be required for clinical purposes in order to establish an individual's hemispheric dominance for language prior to surgery, as in patients with intractable epilepsy. Alternatively, a study may measure an $LI$ to assess the strength or variability in lateralisation for a given language function in order to make inferences about the neural organisation of the language system. That is, studies may vary in whether the aim of $LI$ measurement is to classify or to quantify lateralisation. This will have important implications for which methods of $LI$ calculation are optimal for laterality measurement.

Interpretation of fMRI lateralisation research has been problematic due to a lack of standardisation of fMRI laterality protocols. Multiple arbitrary decisions must be made when calculating the $L$ and $R$ terms for use in the $LI$ equation which might affect the $LI$ value obtained (*Jansen et al., 2006*; *Seghier, 2008*). Such variability in methodology can preclude systematic study of language lateralisation.

For example, when calculating an $LI$ from active voxels in each hemisphere or region of interest (ROI), a decision must be made as to the threshold $p$ value at which to view and analyse the images. Multiple studies have documented the dependence of the $LI$ obtained on the threshold chosen (*Rutten et al., 2002*; *Adcock et al., 2003*; *Seghier et al., 2004*; *Abbott et al., 2010*; *Nadkarni et al., 2015*). As illustrated in Fig. 1, as the threshold value is increased, the number of voxels surviving thresholding decreases, typically leading to an increase in the $LI$. Ultimately, above a certain threshold, no active voxels will remain in the non-dominant hemisphere, resulting in an $LI$ of 1; and below a certain threshold many voxels will survive across both hemispheres, resulting in an $LI$ of 0. Indeed, there are even reports of individuals whose $LI$ shows a switch in dominance with a change in threshold level (*Jansen et al., 2006*; *Suarez et al., 2008*; *Wilke & Lidzba, 2007*; *Ruff et al., 2008*).

This illustrates just one preliminary issue that must be addressed when considering how to quantify lateralisation from fMRI data. Further decisions have to be made as to tasks used in an activation paradigm, whether the analysis focuses on a specific region of interest (ROI) or the whole hemisphere, and whether the quantification of activation is based on magnitude or extent of activity. If the $LI$ is used to categorise individuals as left-, bilateral or right-lateralised, a suitable cut-off for categorisation must also be determined.

The purpose of this review is to assess different protocols for fMRI measurement of language lateralisation used by studies published between 2000 and 2016. We aimed to (1) look at the methods used by different studies over this time period in order to consider whether the field is converging on common criteria for evaluating language lateralisation, and (2) consider evidence for the robustness and reliability of these different methods in order to make recommendations for future research in this field.
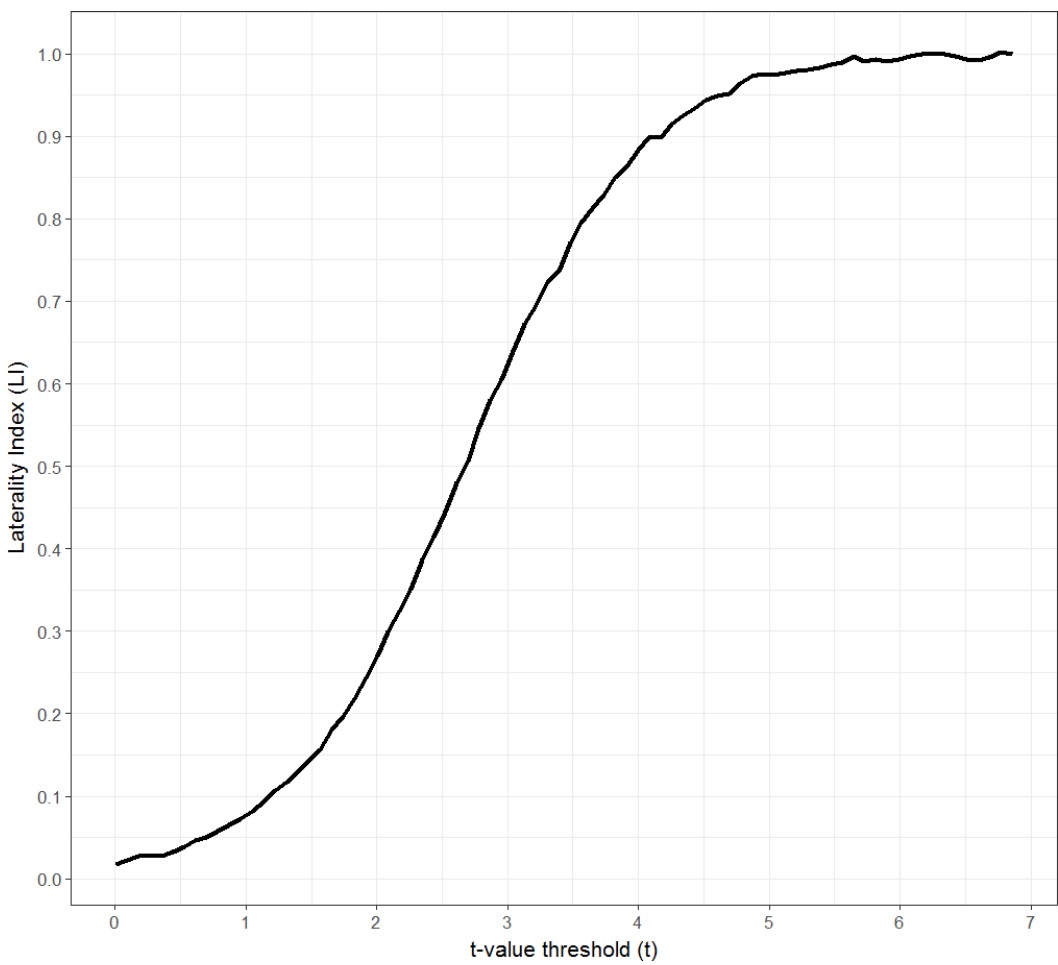

**Figure 1 Threshold dependent laterality curve.** Plot of *LI* as a function of threshold (*t*-value). Figure created by Paul A. Thompson, used with permission.

## MATERIALS AND METHODS

A protocol for this review has been registered on Open Science Framework and can be found at https://osf.io/hyvc4/. This paper addresses those objectives outlined in the protocol relating to assessment of the methods used to quantify lateralisation in fMRI studies of language lateralisation. Assessment of the impact of language task and baselining methods will be considered in a companion paper.

### Eligibility criteria

We reviewed studies of fMRI language lateralisation published between 2000 and 2016. Papers were selected if they met the following inclusion criteria: (1) the paper calculated and reported *LI*s for language using fMRI; (2) participants were healthy monolingual adults; and (3) if participants included both patients and healthy control groups, the data for controls were reported separately. Papers were excluded if: (1) they exclusively studied structural asymmetries, children or bilingualism; or (2) they used language tasks with
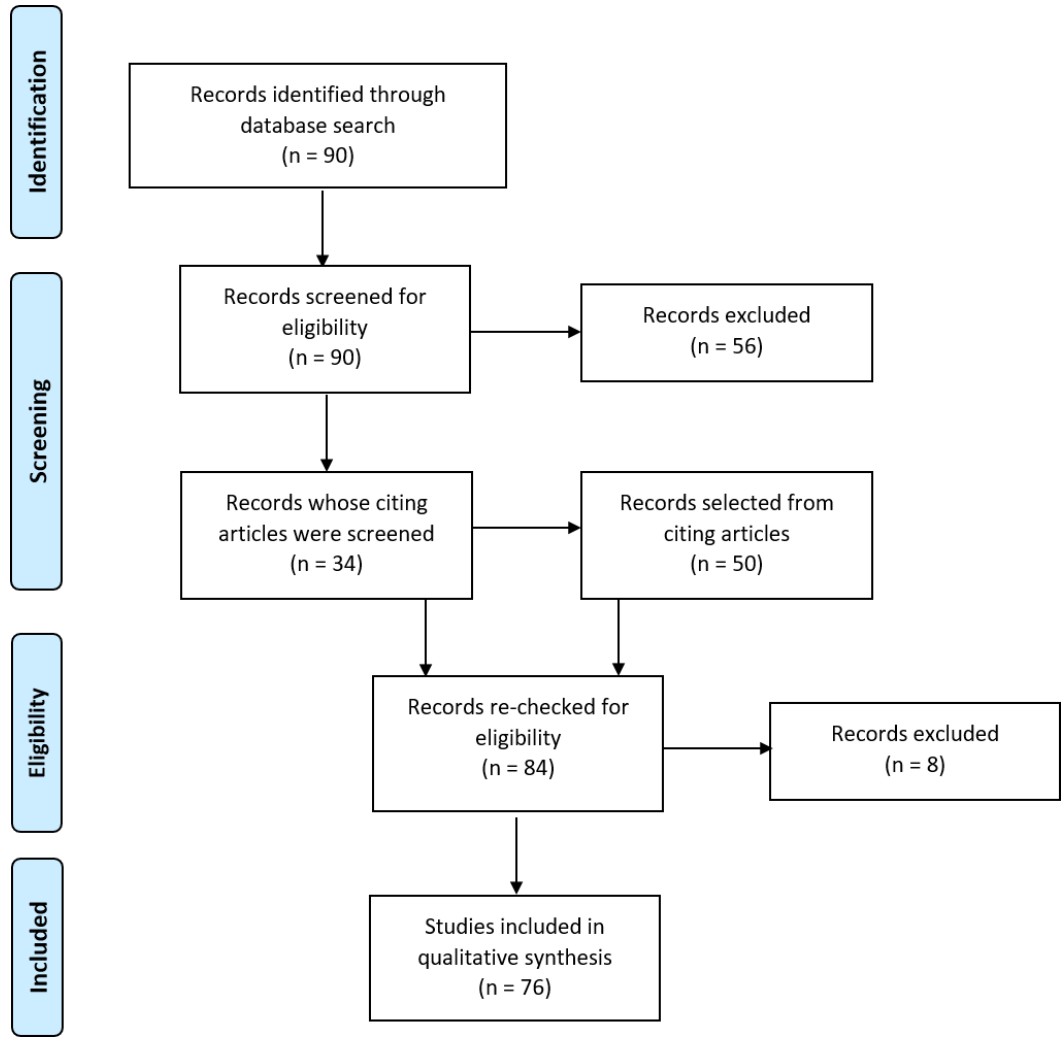

**Figure 2** **Search strategy and selection process.** Flow diagram illustrating the search and selection process for obtaining articles for inclusion in the review. Adapted from *Moher et al. (2009)*.

non-European languages. The rationale for restricting the search to studies on healthy, monolingual, adult participants was to reduce heterogeneity within our study sample.

## Search strategy and selection process

The search and selection process is illustrated in Fig. 2. We searched Web of Science for studies published between 2000 and 2016 using the following search terms: laterali* OR asymmetr* OR dominance; AND language OR reading; AND fMRI OR functional MRI OR functional magnetic resonance imaging OR functional MR OR function MRI; NOT schizophrenia; NOT development*; NOT child*; NOT bilingual*. This was last searched on 05/12/16. Two of the study authors (Abigail Bradshaw and Zoe Woodhead) screened the titles and abstracts of the resulting 90 papers to assess their eligibility then conducted full-text scans to determine whether the inclusion criteria were met. Selected lists were compared between reviewers and any discrepancies discussed and a mutual decision made.

This yielded a total of 34 papers selected from the original 90. To ensure thorough coverage of the literature, papers citing these 34 articles were searched to look for additional articles that met criteria. From this, 50 additional papers were selected, bringing the total to 84 papers. A final search to re-check all 84 papers against search criteria identified 7 ineligible papers. During the review, a further paper was judged to not meet criteria. A list of the final 76 selected papers can be found in Appendix S1.

### Data collection and data summary

For each paper, we recorded the following parameters relating to the protocol used: the type of fMRI design used, the activity measures used for *LI* calculation, the threshold level chosen, the use of global or regional *LI* calculation, the specific regions considered, the language and baseline tasks used, the use of a single or a combined task analysis and the task difficulty. We also recorded sample size and sample handedness for each study. Information on these measures for each paper was collected and managed using REDCap electronic data capture tools (*Harris et al., 2009*) hosted at Oxford University. REDCap (Research Electronic Data Capture) is a secure, web-based application designed to support data capture for research studies, providing: (1) an intuitive interface for validated data entry; (2) audit trails for tracking data manipulation and export procedures; (3) automated export procedures for seamless data downloads to common statistical packages; and (4) procedures for importing data from external sources. The full database can be found in Appendix S2 . A summary table drawn from this database with the key outcomes of interest for this paper is provided in Appendix S3.

The variable nature of the methods used and measures reported by different fMRI studies of language lateralisation means the data are not suitable for a meta-analysis. Instead, this review will document the range of methods used, and provide a qualitative summary of information from these studies that is relevant for our understanding of the robustness and reliability of *LI* measurement.

## RESULTS

### Methods for calculating *LI*

As shown in Fig. 3, a range of methods have been used to compute a laterality index in the set of studies that we considered. These will be briefly described before moving to consider the relative advantages and disadvantages of each. Note that the majority of studies used the standard *LI* ratio approach using the *LI* formula as previously outlined (55 within our search), but use of bootstrapping approaches started to be seen in 2010 and has gained in popularity since that time. Use of different methods likely reflects their availability as well as their robustness; for example, the development of *LI*-tool (*Wilke & Lidzba, 2007*), a tool-box for *LI* calculation within MATLAB software (Mathworks, Natick, MA, USA) has made bootstrapping readily available and easily implementable. In contrast, other approaches such as the flip method and $t$-weighting may be less widely used due to the lack of dedicated software for their implementation. Relatively few studies (3) explicitly compared different methods for calculating the *LI*: we will cover those that did so in more detail in the following sections.

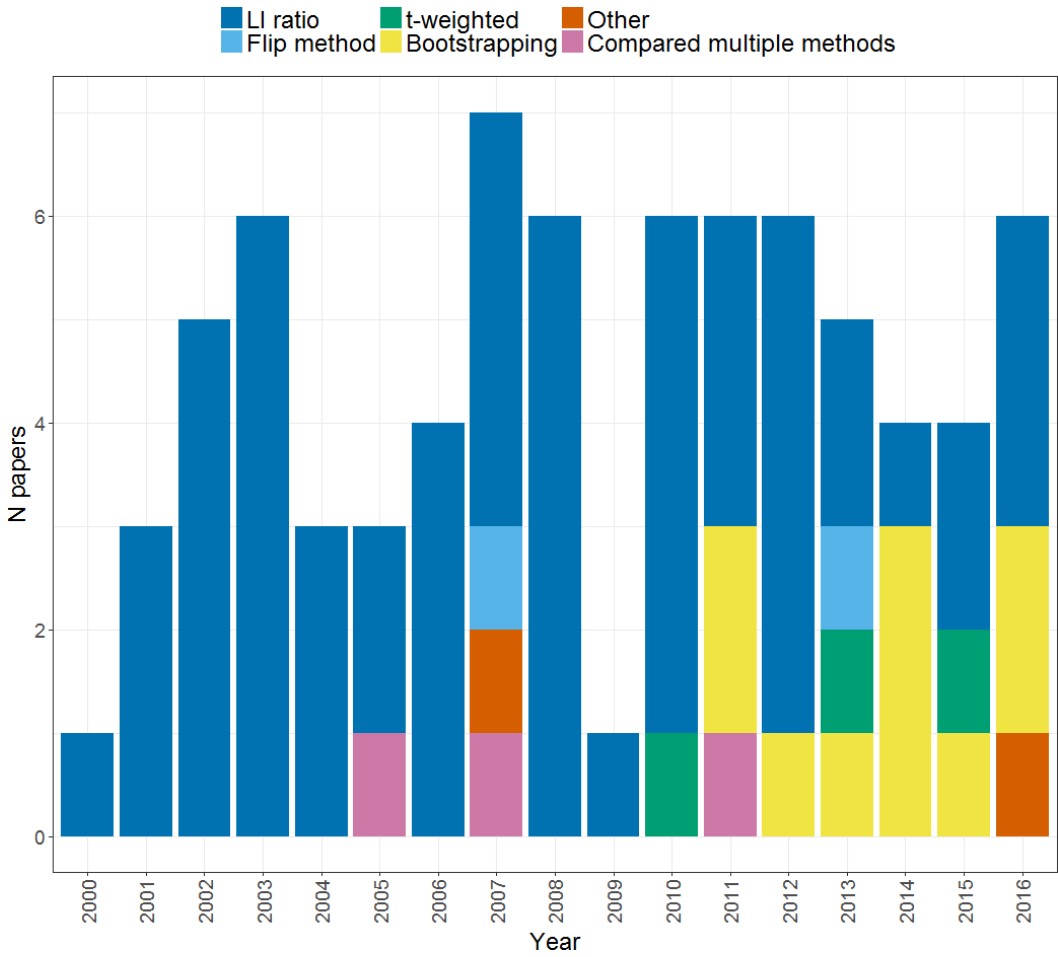

**Figure 3  Methods of calculating an *LI*.** Plot shows the frequency of papers within our search using each method of *LI* calculation across the years from 2000 to 2016.

## Thresholding

Figure 4 illustrates the different thresholding approaches used by studies, and how these have changed over time. As can be seen, the majority of studies (33) used a single fixed threshold approach for determining the *LI* in which a single threshold level is chosen at which either extent of activation, or magnitude of activation in a given region is measured for left and right, and entered into the *LI* formula. As noted above however, use of a single threshold when calculating an *LI* is likely to yield an unreliable and inadequate measure of an individual's pattern of laterality. Awareness of this has led to a decline of the single fixed threshold approach in more recent years, in favour of approaches that aim to address the problem of threshold dependence. Each of these will be described and evaluated in turn in the following sections.

### *Multiple thresholds and threshold dependent laterality curves*

One way to address the problem of threshold dependence is to calculate the *LI* across multiple thresholds. One can then produce a plot of *LI* as a function of threshold (see

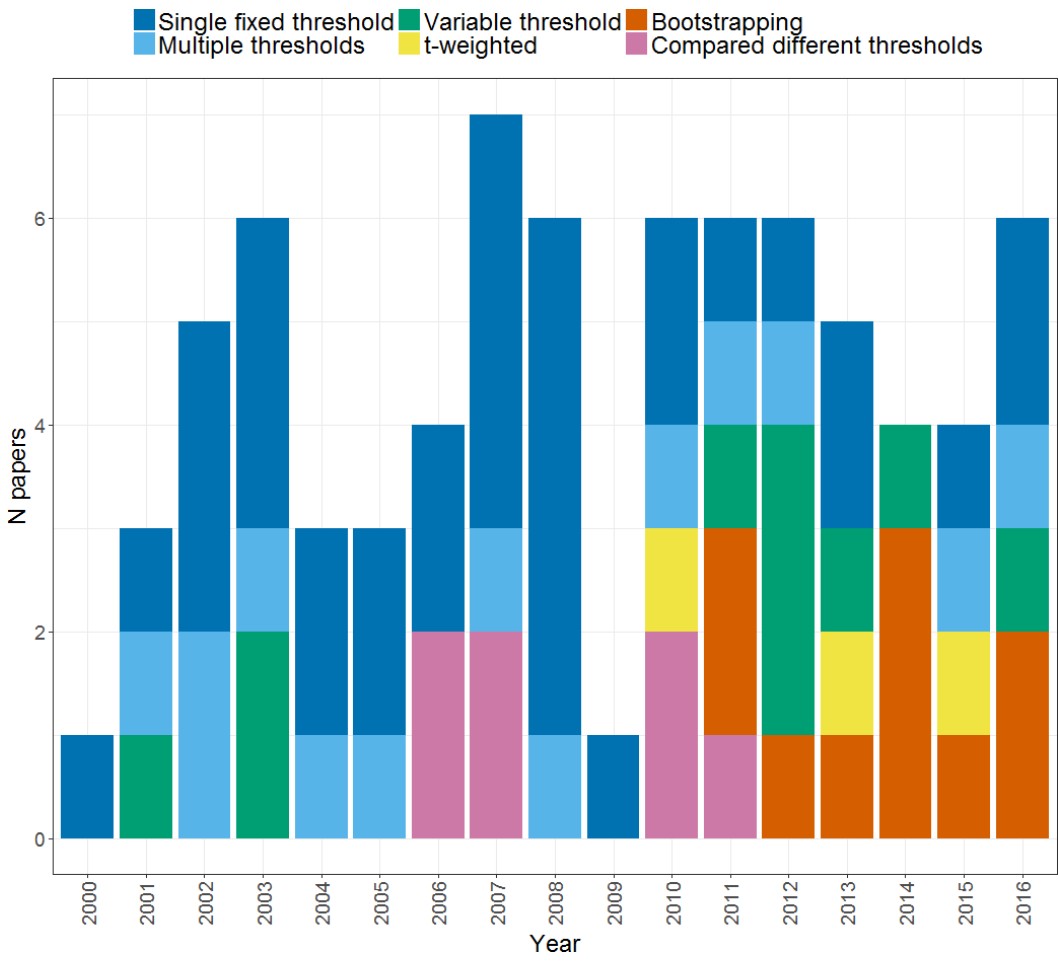

**Figure 4** **Thresholding methods.** Plot shows the frequency of papers using each method of thresholding when calculating an *LI* across the years from 2000 to 2016.

Fig. 1), also known as threshold dependent laterality curves (*Seghier et al., 2004*; *Jansen et al., 2006*; *Ruff et al., 2008*; *Suarez et al., 2008*; *Abbott et al., 2010*). Such curves can allow one to decipher an individual's general tendency towards one pattern of dominance, often showing a transition point at which the increase in laterality plateaus at a particular laterality level. However, such curves are not always informative, since in some cases they may fail to reach a plateau, or are not reproducible within a subject (*Rutten et al., 2002*; *Jansen et al., 2006*).

### Variable thresholds

A second approach uses a variable or adaptive threshold, in which the threshold is set according to subject-specific parameters. One such method involves choosing that threshold which yields a fixed number of active voxels for each individual participant (*Knecht et al., 2003*; *Jansen et al., 2006*; *Abbott et al., 2010*; *Fesl et al., 2010*). Using simulated data, *Abbott et al. (2010)* demonstrated that thresholding at a fixed number of voxels was more robust against variability in signal strength than the standard thresholding method. They advocated

plotting the *LI* as a function of the number of active voxels, similar to threshold dependent curves; these curves are however tighter and more stable. Furthermore, *Fesl et al. (2010)* reported improved reliability of *LI* measurement when using this variable threshold method as opposed to a single fixed threshold. However, this approach does not remove the need for arbitrary decisions, since a 'reasonable' fixed number of active voxels must be decided on. Interestingly, when using this method *Jansen et al. (2006)* set the criterion number of activated voxels at a different level for each language task; this can thus enable one to take into account the fact that different tasks may require different threshold levels.

Other adaptive thresholding methods are based on setting the threshold in proportion to the maximum or mean intensity of voxels in an image. Methods include identifying the highest 5% of voxels with the highest *t*-values, and setting the threshold at half of their mean value (*Fernandez et al., 2001*; *Van Veelen et al., 2011*); alternatively, the mean intensity of voxels within an area of interest can be used (*Stippich et al., 2003*; *Wilke & Lidzba, 2007*; *Partovi et al., 2012a*; *Partovi et al., 2012b*; *Allendorfer et al., 2016*). Using this latter method, *Wilke & Lidzba (2007)* reported more stable *LI*s using variable threshold methods as compared to fixed thresholding, suggesting that this may make *LI*s more robust. This study also demonstrated a flattening of laterality curves when only those voxels that formed a significant cluster or that had a sufficiently low level of variability were included in the *LI* calculation. These clustering and variance weighting methods thus allow calculation of *LI*s to become more stable across threshold levels.

### T-weighting and threshold independent methods

Alternatively, the issue of threshold dependence can be avoided by the use of a 'threshold-independent' method. One such widely used threshold-independent method is *t*-weighting (*Branco et al., 2006*; *Suarez et al., 2008*; *Propper et al., 2010*; *Zaca, Jarso & Pillai, 2013*), illustrated in Fig. 5. This involves plotting a histogram for each hemisphere of the number of active voxels against *t*-score threshold, and then multiplying this distribution by a weighting function that assigns weight in a way directly proportional to *t*-score. The integrated areas under each hemisphere's curve can then be used as the input for the standard *LI* equation. *Suarez et al. (2008)* reported that such a method yielded reduced within-subject and between-subject variability in *LI* compared to fixed thresholding, resulting in clear left lateralisation across subjects.

Other threshold-independent methods have been developed, but these have been less widely used. *Harrington, Buonocore & Farias (2006)* reported that taking the average signal magnitude within activated voxels across multiple thresholds yielded higher and more reproducible *LI*s than a single threshold approach. *Seghier et al. (2011b)* used a method developed by *Nagata et al. (2001)* in which the *L* and *R* terms are calculated by taking the regression of the curve obtained by plotting the number of activated voxels against threshold for each hemisphere separately. This provides a fixed term for each hemisphere that is independent of threshold for use in the *LI* calculation, providing a more robust measure.
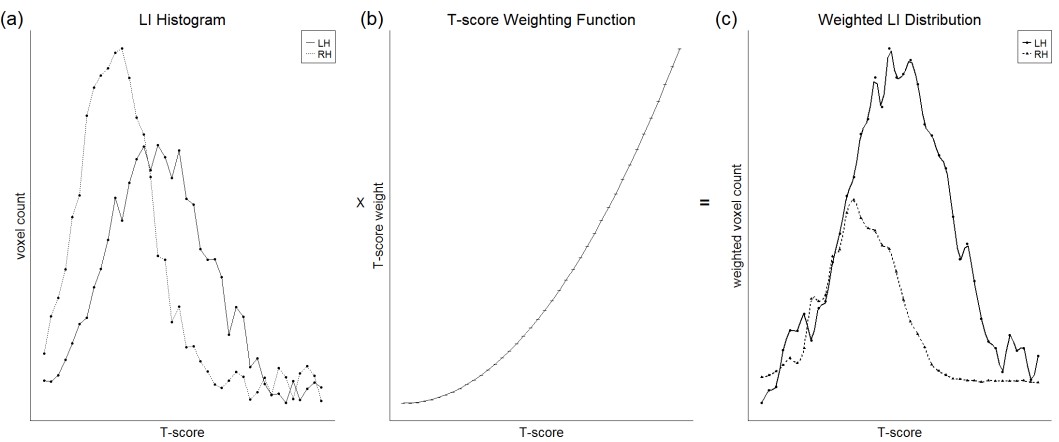

**Figure 5** **Illustration of the *t*-weighting method.** A plot of voxel count as a function of *t*-score threshold (A) is multiplied with a weighting function (B) in which higher thresholds are given greater weight, to obtain a weighted distribution (C). The integrated areas under the right and left hemisphere curves can then be used for the standard *LI* equation. Figure created by Paul A. Thompson, used with permission.

### Bootstrapping

A further method developed to remove the issue of threshold dependence is bootstrapping (*Wilke & Schmithorst, 2006*). This involves iterative resampling and calculation of *LI*s across multiple threshold levels, illustrated in Fig. 6. At each threshold, a vector containing all voxel values is created from an image (b), one for each hemisphere. Multiple random samples of values (e.g., 100 samples) from these vectors are then taken (c) and an *LI* calculated for all possible right/left sample combinations (d). All *LI*s are then plotted in a histogram (e) and a trimmed mean is taken by selecting the central 50% of the data in order to reduce the effect of outliers. A weighted overall mean is then calculated from this resulting data by assigning a higher weight to higher thresholds. This method has been widely adopted in recent research on measuring language lateralisation (*Häberling, Badzakova-Trajkov & Corballis, 2011*; *Van der Haegen et al., 2011*; *Van der Haegen, Cai & Brysbaert, 2012*; *Perlaki et al., 2013*; *Berl et al., 2014*; *Mazoyer et al., 2014*; *Miro et al., 2014*; *Tzourio-Mazoyer et al., 2015*; *Häberling, Steinemann & Corballis, 2016*; *Sepeta et al., 2016*). As well as being threshold-independent, its key strengths include greater resistance to outliers and built-in markers for detecting the presence of outliers within the process of *LI* calculation (*Wilke & Schmithorst, 2006*), making it a robust method for assessing language laterality from fMRI data.

### Activity measure

A key decision in calculation of a laterality index from fMRI data concerns which activity measure to use; signal extent (i.e., the number of suprathreshold voxels in each hemisphere) or signal magnitude (i.e., the average intensity of suprathreshold voxels in each hemisphere). Figure 7 documents the different activity measures used by studies within our search. It can be seen that the majority of studies opt for an extent measure, although in recent years there has been an increased use of magnitude measures.

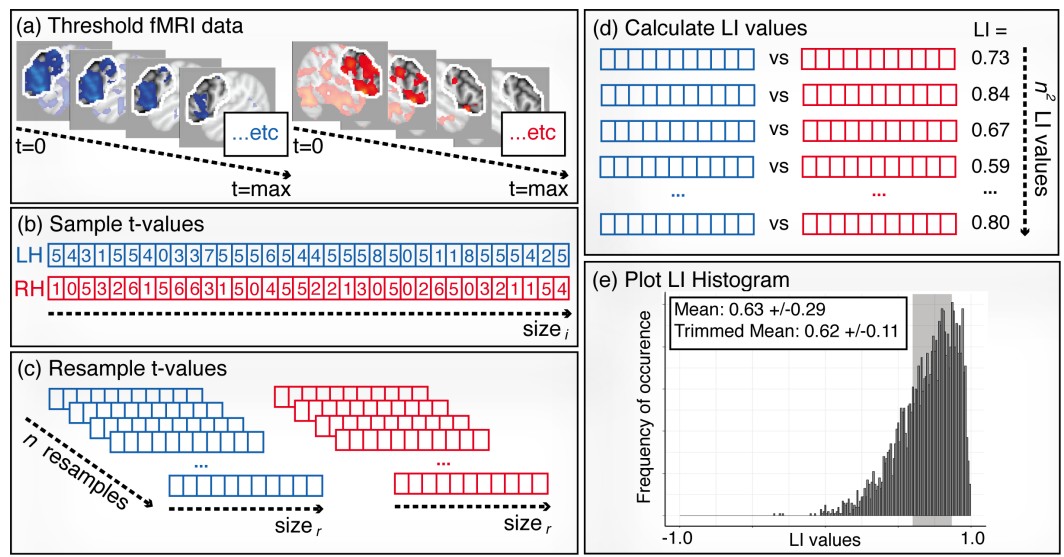

**Figure 6** **Illustration of the bootstrapping method.** (A) Thresholding: contrast images are created across a range of thresholds from 0 to the maximum $t$-value. (B) Sampling: for each threshold level, a sample of $t$-values (size $_i$) are randomly selected from the left and right ROIs . (C) Resampling: values from the sample vectors are randomly resampled $n$ times, each with size $_r$. (D) LI calculation: LI values are calculated for all possible combinations of right and left resamples, creating $n^2$ LI values in total. (E) Histogram: steps (B)–(C) are repeated for all threshold levels, and all of the resulting LI values are plotted in one histogram. A trimmed mean, taken from the middle 50% of the data (shaded area), is used as the final LI measure.

Of the studies that compare both methods, many have reported finding similar laterality indices and curves (*Jansen et al., 2006*; *Bethmann et al., 2007*; *Wilke & Lidzba, 2007*; *Ocklenburg, Hugdahl & Westerhausen, 2013*). Others have reported differences in *LI* strength, with reports of both higher *LI*s for magnitude measures (*Harrington, Buonocore & Farias, 2006*) and higher *LI*s for extent measures (*Jensen-Kondering et al., 2012*). Further still, *Jansen et al. (2006)* reported that differences in the activity measure used for calculating laterality with a picture naming task could yield different dominance classifications for a given participant. This was not the case however for verbal fluency and semantic decision tasks, suggesting that this may reflect something particular about the activity patterns induced by naming.

There is evidence that these measures can yield the same high levels of reproducibility for *LI* measurement: *Morrison et al. (2016)* reported 100% reproducibility for classification of language dominance and almost identical test-retest *LI* correlations using both activity measures for a rhyming task. However, the majority of studies report higher reproducibility for signal magnitude measures compared to signal extent measures (*Adcock et al., 2003*; *Jansen et al., 2006*; *Harrington, Buonocore & Farias, 2006*; *Morrison et al., 2016*). Importantly, *Jansen et al. (2006)* reported that a magnitude measure determined dominance reproducibly only when just those voxels that exceeded a criterion activation level were included; an extent measure was not reproducible. Magnitude measures have also been reported to be less sensitive to noise than a thresholded extent measure (*Adcock et al., 2003*).
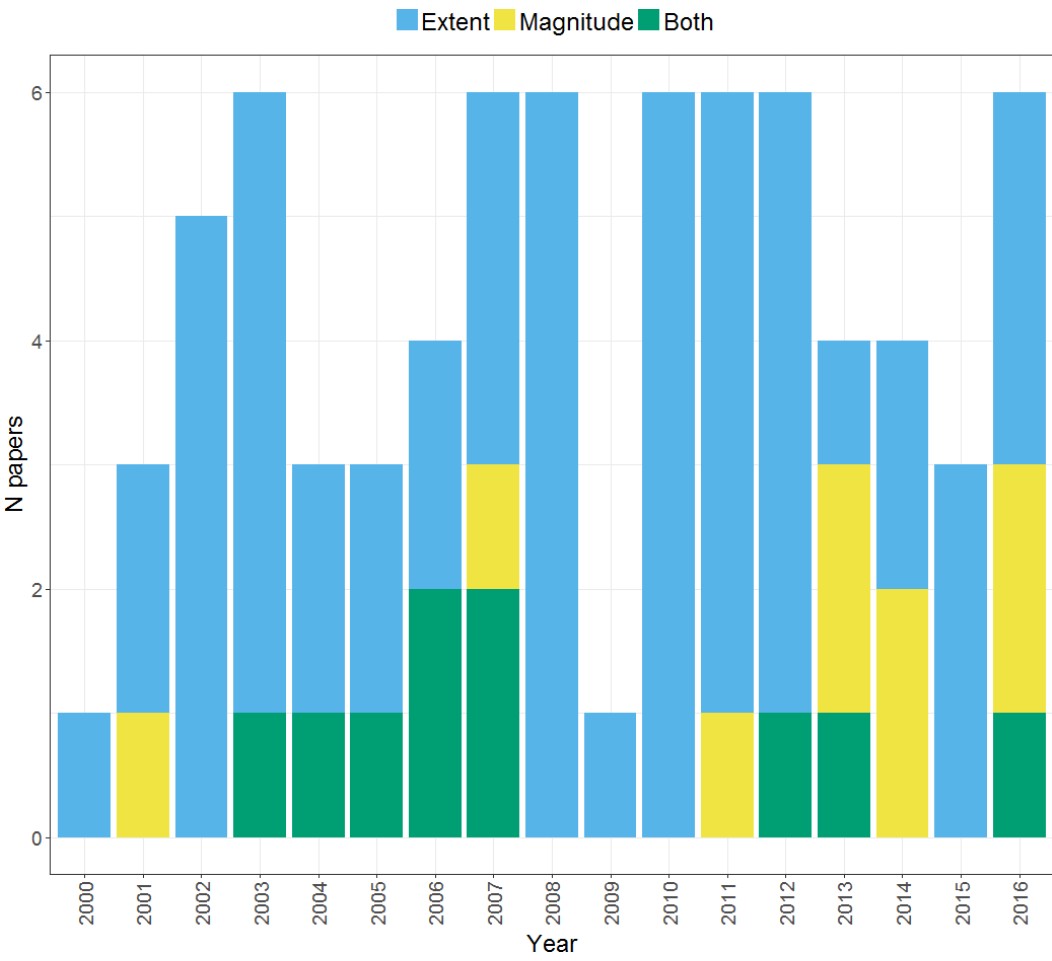

**Figure 7** **Activation measure used for *LI* calculation.** Plot shows the frequency of papers within our search using each type of activation measure across the years from 2000 to 2016.

One possible explanation for higher reliability with magnitude than extent *LI*s relates to intersessional differences in the distribution of *t*-values. *Gorgolewski et al. (2012)* discuss the phenomenon in which shifts in *t*-value histograms can occur between sessions due to global effects which have a systematic effect across the whole brain. This can result in the distribution of *t*-values in a second session being shifted relative to the first, whilst the pattern or shape of that distribution is still preserved. In such cases, use of the same fixed threshold across both testing sessions could result in greatly different activity maps, leading to a low estimation of reliability, despite that fact that the overall pattern has not changed. Extent measures would be affected by such shifts to a greater extent than magnitude measures, thus resulting in their reduced reliability.

Of further note is *Jansen et al.*'s (*2006*) finding that *LI*s based on signal extent lacked meaningful variation, often yielding *LI* values of 1; in contrast, *LI* magnitude measures gave greater between-subject variation in *LI* values. Which constitutes better laterality data depends on one's view of lateralisation measurement. That is, in cases when one wishes to classify individuals' language dominance, having *LI* values close to 1 or −1

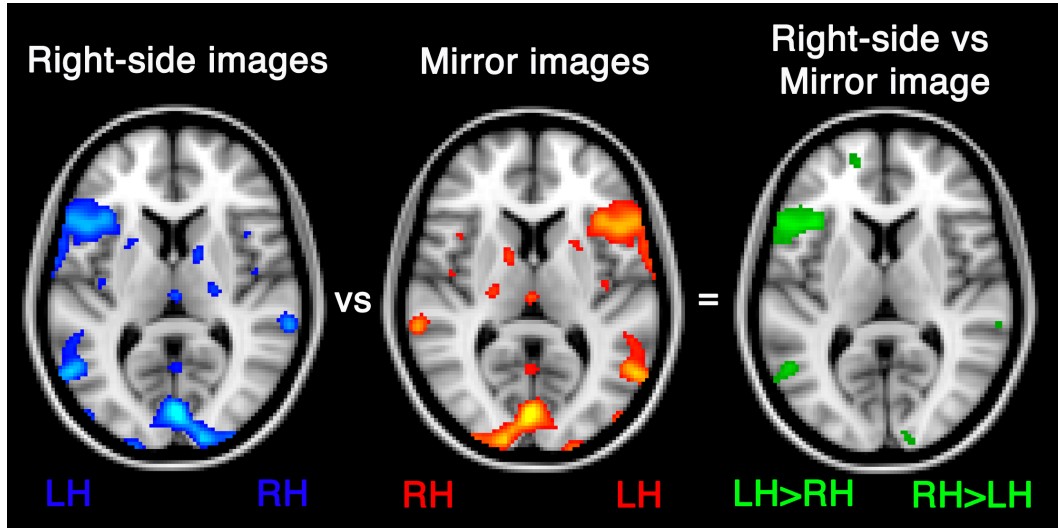

**Figure 8 The flip method.** By contrasting a right-side contrast image with a mirror image (flipped so that the right hemisphere is on the left), a new contrast image is generated with significant voxels indicating regions in which left activity is statistically significantly greater than right homologue activity.

would be useful to allow decisions to be clear cut. Conversely, when one is interested in quantifying individual variability in the degree of language lateralisation beyond the binary typical/atypical distinction, an *LI* measure that reveals greater between-subject variation would be more useful.

### The flip method

The standard *LI* ratio does not indicate whether the difference in activity between the hemispheres is significant or not, but simply quantifies the bias in activity towards one hemisphere. The flip method (*Baciu et al., 2005*) was developed to provide a direct statistical comparison of activity between the hemispheres. As illustrated in Fig. 8, this involves contrasting two sets of functional images created for the contrast of interest (i.e., task versus control); a right side images set, in which the left hemisphere is on the left and a mirror images set, in which the image is flipped such that the left hemisphere is on the right. By contrasting these two images, one can identify those homotopic voxels that show a significant difference in activity across the hemispheres. The resulting significant voxels in each hemisphere can then be used as input for the standard *LI* equation.

This method has been used in a small number of studies (see Fig. 3) as a means of measuring laterality (*Cousin et al., 2007*; *Seghier et al., 2011b*; *Hernandez et al., 2013*). *Baciu et al. (2005)* compared the flip method to the standard *LI* ratio by comparing the correlations of each method's *LI*s with handedness lateralisation indices. While both methods yielded language *LI*s that were poorly correlated with handedness *LI*s for a verb generation task, when a rhyming task was used the flip method yielded higher correlations than the standard *LI* method. Such a finding is difficult to interpret, especially given the inconsistent relationship between handedness and language dominance (e.g., *Mazoyer et al., 2014*).

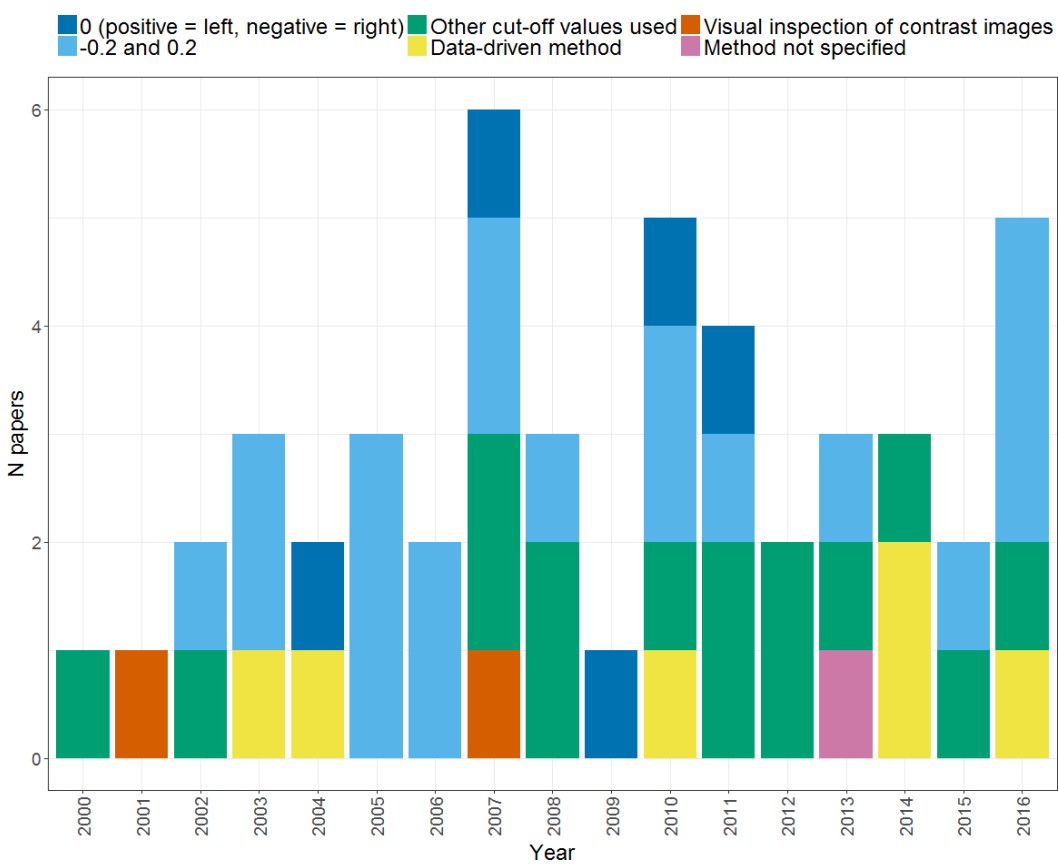

**Figure 9 Methods of dominance classification.** Plot shows the different methods of classifying language dominance used by studies within our search across the period from 2000 to 2016. Note that studies within our search which did not classify dominance are not included in this plot.

It should be noted that the flip method cannot be used to describe bilaterality, but only non-significantly lateralised activity. Further, since significant voxels from this contrast typically reside exclusively in the dominant hemisphere, using these voxels as input for the *LI* equation will often yield an *LI* of −/+1, rendering it an inappropriate method to provide a meaningful description of degree of lateralisation.

## Dominance classification

If a categorical dominance classification is required, some form of standardised procedure is needed once an *LI* has been calculated. The range of methods used by different studies within our search is illustrated in Fig. 9. The most standard method of dominance classification uses cut-offs at −0.2 and 0.2, to divide left dominance ($LI > 0.2$) from bilaterality ($-0.2 \leq LI \leq 0.2$) and right dominance ($LI < -0.2$). However, such cut-offs are arbitrary, and we found multiple studies within our search that chose their own cut-offs (see 'other cut-off values used'), including 0.1, 0.33, 0.4, 0.5 and 0.6. Thus, it can be seen that there is a high level of heterogeneity in the methods of dominance classification used by different studies. This makes it very difficult to draw conclusions and comparisons between the proportions of typically and atypically lateralised individuals reported by

different studies, and thus impedes progress in understanding the distribution of such lateralisation profiles across different populations.

Other researchers have investigated data-driven methods of defining dominance categories, in which the typicality/atypicality of cases is assessed in relation to norms within the data itself or a separate 'normative' dataset (*Adcock et al., 2003*; *Seghier et al., 2004*; *Abbott et al., 2010*; *Berl et al., 2014*; *Mazoyer et al., 2014*; *Tzourio-Mazoyer et al., 2016*). One simple way of deriving cut-offs is to use 2 standard deviations below the mean as a threshold to divide typical from atypical laterality (*Adcock et al., 2003*; *Seghier et al., 2004*). *Abbott et al. (2010)* used a similar approach in which an individual's *LI* distribution (*LI* as a function of voxel count) was compared to a normative distribution based on a sample of controls with 'typical' lateralisation. As illustrated in Fig. 10, the individual's threshold-dependent laterality curve is established as being either within or below the lower 95% confidence interval for the control group (represented by the shaded area in Fig. 10). If an individual's laterality fell below such an interval, they were classed as atypical, on the basis of a low probability of their laterality data having come from the 'population' of the normative group. This method thus provides an objective means of establishing an individual as either atypical or typical in relation to a normative group.

Other data-driven approaches to dominance classification suggest the existence of more than two dominance categories. *Berl et al. (2014)* used a hierarchical clustering method which gradually and iteratively combined cases into clusters and indicated at what level in the hierarchy the optimal cluster solution was obtained. They discussed both a three cluster solution and a two cluster solution for a large sample of right handers; the former divided subjects into left dominant, crossed dominance (one outlying case) or bilateral, whereas the latter divided typically and atypically lateralised subjects. This latter solution was found to divide at an *LI* of 0.5, which was therefore argued to represent a meaningful cut-off for dominance classification.

A larger-scale study by *Mazoyer et al. (2014)* looked at dominance categories in both left and right handed participant groups. They used Gaussian mixture modelling to extract dominance categories from laterality data (consisting of *LI* values between $-100$ and $+100$), which involves determining the optimal number of Gaussian functions that can be fitted to the data. This found different model solutions for right and left handed groups. For the right handed group (Fig. 11A), a three function solution was optimal, consisting of two overlapping 'typical' (left dominant) functions (both with *LI* values above $+18$) and a third 'ambilateral' function (consisting of *LI* values between $-50$ and $+18$). This agrees with *Berl et al.*'s (*2014*) study, which found no evidence for right hemisphere dominance in a right handed group. However, in the left handed group (Fig. 11B), an additional 'strongly atypical' function was found with strongly negative *LI* values (below $-50$). Only 10 left handers from this large sample (297 right and left handers) were strongly atypical, indicating that it is very rare. They thus argued for the need to treat atypical laterality as a heterogeneous group consisting of the subgroups of 'ambilateral' and 'strongly atypical'; conversely, it was argued that the overlapping typical distributions (from both left and right handed participants) could be combined into a single homogenous typical group.

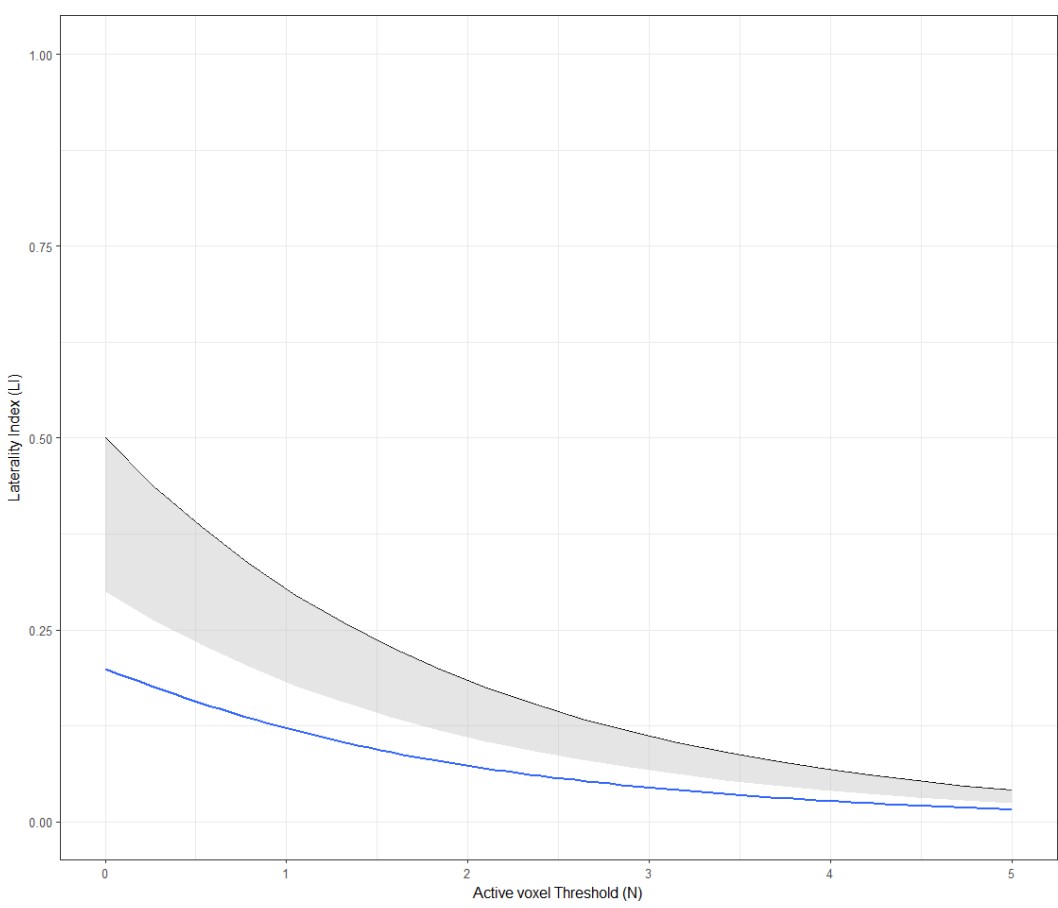

**Figure 10 *Abbott et al.*'s (*2010*) method of dominance classification.** The laterality curve of the subject (blue) is compared to that of a normative control group (black), using the lower 95% confidence interval for the control group (represented by the shaded area). Figure created by Paul A. Thompson, used with permission.

These data-driven methods of deriving dominance categories suggest that the traditional −0.2 and 0.2 cut-offs do not reflect the true distribution of *LI*s across individuals, and that revision of standards for dominance classification is needed in the field, with ramifications for both clinical practice and research. It should be further noted that classification of bilaterality from fMRI data is typically not reproducible across repeated testing sessions or even across different methods of *LI* calculation (*Jansen et al., 2006*), suggesting the need to interpret bilateral classification with caution. Further work is needed to establish the reproducibility of classification into the 'ambilateral' group identified by *Mazoyer et al. (2014)*.

### Effect of region of interest on laterality

Another key consideration in calculating an *LI* concerns whether to include all voxels across a hemisphere (to calculate a global *LI*) or whether to define a region or several regions of interest (a regional *LI*). Figure 12 shows changes in approach to choice of region over time across the studies within our search. It can be seen that the majority

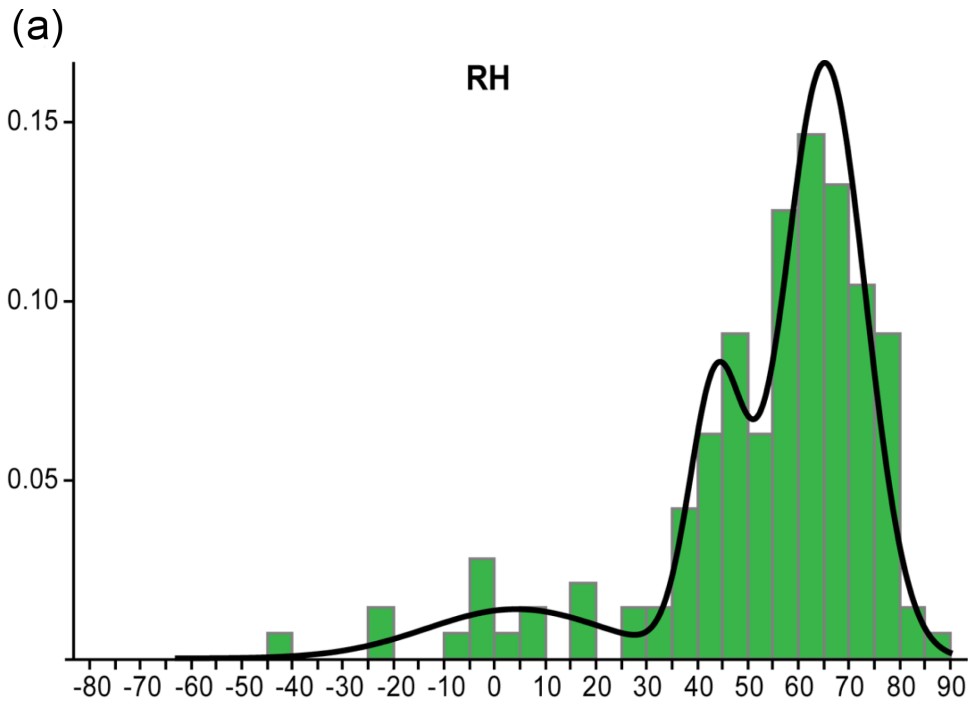

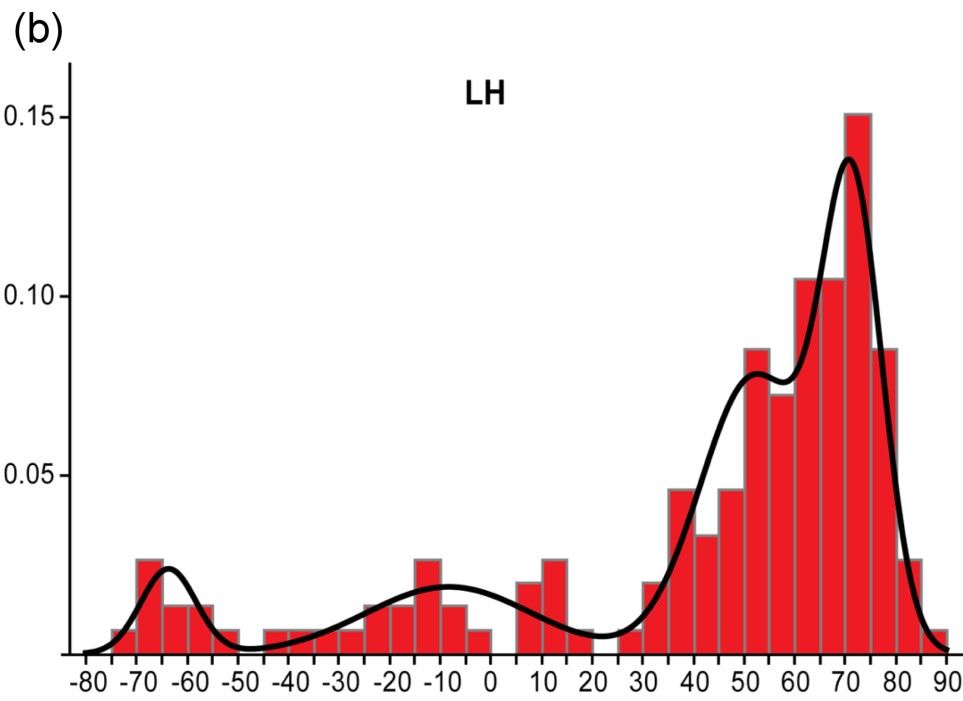

**Figure 11** *Mazoyer et al.'s (2014)* **method of dominance classification.** Histograms showing the distribution of *LI* values across samples of right handed (A) and left handed (B) individuals, with the envelope showing the Gaussian functions fitted to the data for determination of dominance groups. Reprinted from *Mazoyer et al. (2014)*.

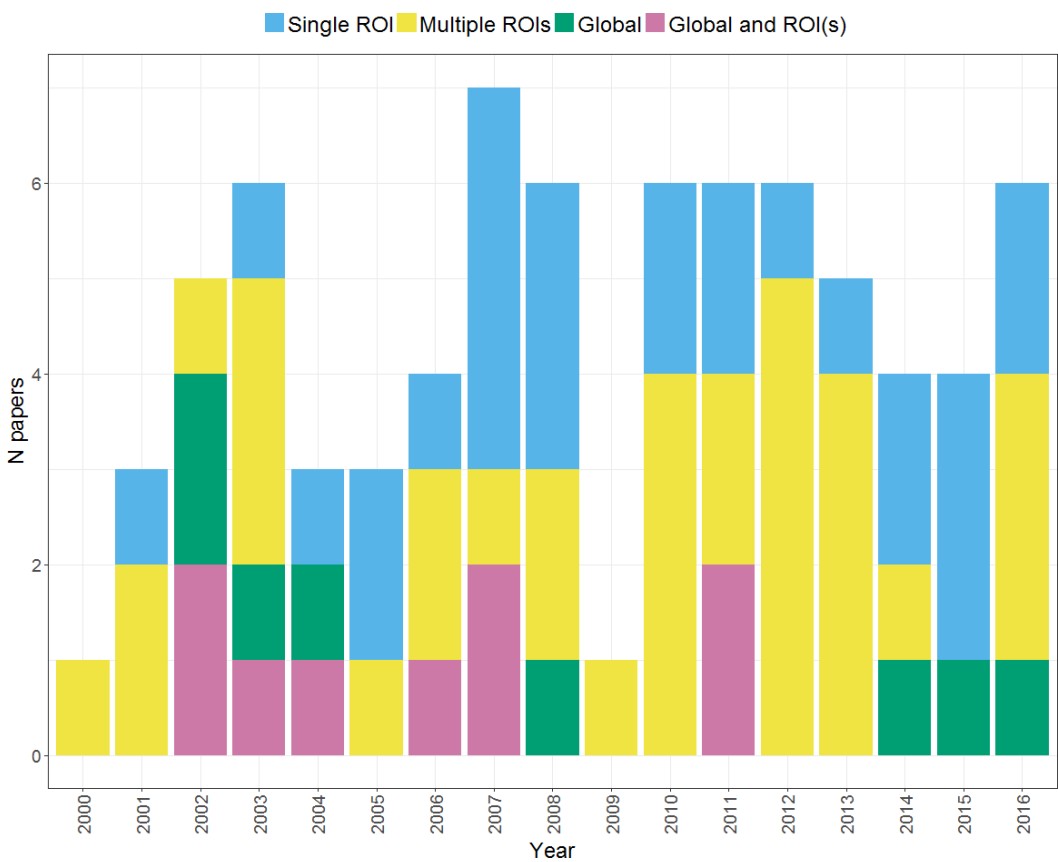

**Figure 12  Use of global and regional approaches to *LI* calculation over time.** Plot shows the regional approaches used for *LI* calculation by studies within our search across the period from 2000 to 2016.

of studies use a regional *LI*, either picking a single ROI or calculating multiple *LI*s from different regions. A number of studies have compared global and regional approaches to *LI* calculation, particularly in earlier years within our search period. This evidence on the relative advantages and disadvantages of the global and regional approaches is discussed below. It can also be seen that a substantial proportion of studies use a single ROI in *LI* calculation; however, a review of evidence on regional variability in laterality questions the adequacy of such an approach (see *Regional heterogeneity in laterality*).

### *Global and regional LIs*

A potential issue with global *LI*s is that voxels outside the areas most relevant to the language paradigm can have a strong influence on the *LI* obtained. Whilst many studies have shown that *LI*s from pre-specified language ROIs are stronger and more reliable than global *LI* measurements (*Fernandez et al., 2001*; *Rutten et al., 2002*; *Suarez et al., 2008*; *Pravata et al., 2011*), others have found no difference (*Hund-Georgiadis et al., 2002*) or the opposite (*Rutten et al., 2002*; *Wilke & Lidzba, 2007*). Further studies suggest that whether global or regional *LI* measures are optimal may depend on other methodological decisions. For example, *Jansen et al. (2006)* reported that global *LI*s were more reliable than regional *LI*s when a voxel count approach was used. Similarly, *Rutten et al. (2002)* reported greater

reliability for global *LI*s over regional *LI*s during an antonym generation task, but that the reverse was true for a verb generation task.

In general however, cases of crossed dominance or regional heterogeneity in lateralisation have been used to argue for the need for regional rather than global laterality indices, which fail to capture these finer grained individual patterns in regional laterality (e.g., *Seghier et al., 2011a*; *Seghier et al., 2011b*). Such evidence will be discussed in the following section, *Regional heterogeneity in laterality*.

### Regional approaches to LI calculation

When using a regional rather than a global *LI*, careful thought must be given to deciding which areas to choose as ROIs. Typically, areas within frontal and temporoparietal cortex are chosen for measuring language laterality, such as the inferior frontal cortex or posterior superior temporal gyrus. It is not the case that any single ROI or combination of ROIs will always be optimal for assessing laterality; instead, the choice of ROI(s) must be guided by other factors such as the language function being studied or the purpose of laterality measurement.

There are a number of studies which measure laterality from both frontal and temporoparietal ROIs, which allows one to compare the robustness and reliability of their *LI*s. There is mixed evidence over whether frontal or temporoparietal ROIs yield stronger or more reliable *LI* values. The majority of studies report stronger laterality in frontal than temporoparietal ROIs, across a wide range of both expressive and receptive tasks (*Vikingstad et al., 2000*; *Gaillard et al., 2003*; *Clements et al., 2006*; *Harrington, Buonocore & Farias, 2006*; *Vernooij et al., 2007*; *Szaflarski et al., 2008*; *Niskanen et al., 2012*; *Partovi et al., 2012a*; *Partovi et al., 2012b*; *Propper et al., 2010*; *Ocklenburg, Hugdahl & Westerhausen, 2013*). However, some have reported the opposite, a pattern particularly associated with the use of receptive tasks such as semantic decision, speech listening or auditory comprehension (*Fernandez et al., 2001*; *Ramsey et al., 2001*; *Hund-Georgiadis et al., 2002*; *Harrington, Buonocore & Farias, 2006*; *Bethmann et al., 2007*; *Brennan et al., 2007*; *Sanjuan et al., 2010*; *Van Oers et al., 2010*; *Jensen-Kondering et al., 2012*; *Niskanen et al., 2012*; *Häberling, Steinemann & Corballis, 2016*). A similar pattern emerges for reliability of laterality measurement; while frontal *LI*s are often reported as more reliable than temporoparietal *LI*s (*Harrington, Buonocore & Farias, 2006*; *Szaflarski et al., 2008*; *Partovi et al., 2012a*), the reverse can be true when using a receptive language task (*Harrington, Buonocore & Farias, 2006*; *Jansen et al., 2006*). This suggests that the two areas are both capable of yielding robust and reliable laterality measurement, provided that a receptive task is used to engage temporoparietal areas.

### Regional heterogeneity in laterality

Research comparing laterality across regions has reported cases of crossed or dissociated dominance across different cortical language areas, at both an individual and a group level (*Vikingstad et al., 2000*; *Thivard et al., 2005*; *Jansen et al., 2006*; *Bethmann et al., 2007*; *Propper et al., 2010*; *Seghier et al., 2011a*; *Seghier et al., 2011b*; *Van der Haegen, Cai & Brysbaert, 2012*; *Vingerhoets et al., 2013*; *Berl et al., 2014*; *Häberling, Steinemann & Corballis, 2016*). *Bethmann et al. (2007)* reported four subjects with crossed frontal and

temporal dominance for a semantic decision task; in particular, one subject was classified as bilateral when ROIs were combined, whereas classification based on either only a frontal or a temporal ROI yielded contradicting dominance categories. Other studies report within-subject dissociations between the dominance measured from different regions using different language tasks. Cases of crossed dominance have been reported between frontal *LI*s for expressive language and temporal or occipitotemporal *LI*s for receptive language (*Van der Haegen, Cai & Brysbaert, 2012*; *Häberling, Steinemann & Corballis, 2016*), and between temporal and frontal *LI*s for two different expressive language tasks (*Vikingstad et al., 2000*). In *Van der Haegen, Cai & Brysbaert*'s (*2012*) study, the majority of left handed subjects showed colateralisation of inferior frontal cortex for verb generation and ventral occipitotemporal cortex for lexical decision, but a small minority (3 out of 57 participants) showed crossed dominance. Such crossed dominance is not limited to left handers however; *Häberling, Steinemann & Corballis (2016)* reported cases of crossed frontal expressive and temporal receptive dominance among both left and right handers.

Such cases highlight the inadequacy of relying on a single global or regional *LI*, and how this practice in previous research may have led to an underappreciation of such crossed dominance in the literature. Indeed, such regional variation is not only found at the individual level, but also at a group level. In a comparison of lateralisation for a semantic word matching task across 50 different ROIs, *Seghier et al. (2011a)* and *Seghier et al. (2011b)* reported a negative correlation between *LI*s obtained from the angular gyrus and the ventral precentral gyrus. This suggests that regional heterogeneity can be a normal part of typical profiles of hemispheric lateralisation for language.

## SUMMARY AND CONCLUSIONS

This review has highlighted the many different ways in which calculation of an *LI* from fMRI data presents a methodological challenge, and how the use of different methods has changed over time. Of course, the decisions one makes when designing an fMRI laterality experiment will depend on the question being investigated; however, here we have highlighted some key principles that emerge from the literature that should be considered in order to generate increased standardisation in fMRI laterality protocols across future studies. Increased homogeny in the methods used by different studies will enable better integration of research findings in order to draw conclusions as to the nature and correlates of language lateralisation.

fMRI *LI* calculation must address the problem of threshold dependence. Bootstrapping represents a promising method for calculating a robust, threshold-independent *LI*, making it a widely used method in recent research. The general pattern of evidence suggests that signal magnitude may provide a more robust and reliable measure than signal extent, and that regional *LI*s calculated from pre-specified ROIs are stronger and more reliable than global *LI*s. However, such decisions need to be considered in light of other methodological parameters (e.g., the activity measure used) in order to optimise the fMRI analysis. A useful tool for implementing such analysis methods is *LI*-tool, a tool-box within MATLAB software (Mathworks, Natick, MA, USA), developed by *Wilke & Lidzba (2007)*. This

includes options for different thresholding techniques and activity measures, and can implement the bootstrapping method.

Data-driven methods can provide a less arbitrary means of classifying language dominance and support the validity of a three category model of language dominance within a mixed handedness sample, consisting of typical (left dominant), ambilateral, and atypical (strongly right dominant) groups; conversely in right handed samples a two-category model (typical versus ambilateral) may be sufficient (*Mazoyer et al., 2014*). No subsequent studies have implemented the thresholds for dominance classification suggested by *Mazoyer et al.*'s (*2014*) large scale study, except a paper reporting on the same sample of right and left handed individuals (*Tzourio-Mazoyer et al., 2016*). Further work is needed to implement and validate these cut-offs, to see if these generalise to other samples and thus could be used as standard practise.

The choice of which regions of interest to use for *LI* calculation again depends on the question being asked. If one wishes to classify laterality for a particular language function, one must consider which ROI yields the highest and most reliable *LI*s for that function. Frontal ROIs typically yield the strongest and most reliable laterality for expressive tasks, whereas a temporoparietal ROI may be more appropriate for receptive tasks. However, measurement of laterality from a single regional or global ROI can be misleading and does not capture potential regional heterogeneity. This was highlighted recently by *Tailby, Abbott & Jackson (2017)* in relation to the need to appreciate such regional variability in presurgical planning with epilepsy patients, and the consequent inadequacy of a single metric to quantify an individual's 'language dominance'. Therefore, in fMRI laterality protocols, lateralisation across frontal and temporoparietal ROIs for at least one expressive and one receptive task should be measured, to obtain a comprehensive picture of any individual's pattern of hemispheric dominance for language. This will enable further work to investigate the significance of such regional heterogeneity in dominance; for example, are there any functional consequences of having crossed frontal-temporal language laterality? In this way, fMRI as a method of laterality measurement can provide unique insights into lateralisation at a regional level; this should be fully exploited in future research.

Lastly, it should be noted that fMRI data provides opportunities for investigating the nature of lateralisation for a function that goes beyond simple calculation of laterality indices. For example, fMRI can be used to investigate more mechanistic questions such as potential relationships between inter- and intrahemispheric connectivity and lateralisation (e.g., *Seghier et al., 2011a*; *Frässle et al., 2016*). In order to pursue more advanced analyses, it will be important for the field to establish robust protocols for laterality measurement as standard across studies.

## ACKNOWLEDGEMENTS

We would like to thank Paul A. Thompson for his help in creating figures for this manuscript. We would also like to thank Cyril Pernet for his feedback and helpful suggestions on a preprint of this manuscript.

### Funding

This work was supported by an Advanced Grant awarded by the European Research Council (project 694189 - Cerebral Asymmetry: New directions in Correlates and Etiology –CANDICE). Dorothy Bishop is funded by a programme grant 082498/Z/07/Z from the Wellcome Trust. There was no additional external funding received for this project. The funders had no role in study design, data collection and analysis, decision to publish, or preparation of the manuscript.

### Grant Disclosures

The following grant information was disclosed by the authors:
European Research Council: 694189.
Wellcome Trust: 082498/Z/07/Z.

### Competing Interests

Dorothy Bishop is an Academic Editor and Academic Advisor for PeerJ.

### Author Contributions

- Abigail R. Bradshaw performed the experiments, analyzed the data, wrote the paper, prepared figures and/or tables.
- Dorothy V.M. Bishop conceived and designed the experiments, prepared figures and/or tables, reviewed drafts of the paper.
- Zoe V.J. Woodhead prepared figures and/or tables, reviewed drafts of the paper.

### Data Availability

    Figshare: https://figshare.com/projects/Methodological_considerations_in_assessment_of_language_lateralisation_with_fMRI_a_systematic_review/20903.
    Github (code): https://github.com/p1981thompson/Candice.

### Supplemental Information

Supplemental information for this article can be found online at http://dx.doi.org/10.7717/peerj.3557#supplemental-information.

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
