# Peer review of "Methodological considerations in assessment of language lateralisation with fMRI: a systematic review"

_PeerJ, doi:10.7717/peerj.3557_

## Round 0.1 · original submission · Minor Revisions

Please address the minor comments provided by the reviewers, both of whom appreciated the manuscript.

·

Basic reporting

Overall, clarity and coverage of existing literature is excellent.

I have one suggestion. The abstract (lines 17-21) identifies the aim of the work by stating that, "the methods used in the fMRI laterality literature currently are highly variable, making systematic comparisons across studies difficult. Here we consider the different methods of quantifying and classifying laterality that have been used in fMRI studies since 2000, with the aim of determining which give the most robust and reliable measurement."
The first paragraph of Intro, however, ends (lines 32-35) "However, our understanding of the nature and correlates of such lateralisation is relatively limited. Many questions remain, such as the functional relevance of such hemispheric specialisation and the significance of individual variation in language dominance." This makes it sound like the main focus of the study is going to be these issues (relevance of hemispheric specialisation, significance of individual variation), however the aim of the paper really seems to be that outlined in the abstract. The question of the relevance and significance of hemispheric specialisation is secondary (from the perspective of the manuscript) to the question of how can we actually quantify and classify laterality. I'd suggest replacing these sentences (lines 32-35) with a statement along the lines of the passage from the abstract quoted above (lines 17-21), in order to better frame the paper.


Also, I think there is a typo at Line 177: I think the authors mean "in turn" rather than "in term".

Experimental design

This is a qualitative review article, and the authors' identification of the problem addressed is clear (variability in calculating LI, review of these methods, identification of strengths and weaknesses of each) methods with respect to identification and inclusion of articles is clear and logical.

Validity of the findings

The only comment I have to make here relates to the notion that having to collect a normative comparison group is a 'limitation'. At lines 378-380, with regard to the discussion concerning classification of atypicality and the Abbott et al (2010) paper, the authors state: "[the Abbott et al] method thus provides an objective definition of atypical lateralisation, but with the obvious limitation that it relies on having a normative comparison group to define ‘typical’ lateralisation".
This isn't actually a limitation as if the objective is to identify what is atypical, then one needs to first identify what is typical. This is the fundamental tenet of all normative testing in health science. Simply defining an LI of 0.2 or less as atypical would not necessarily make it so (as the authors indeed already state, this is just an arbitrary cutoff). Atypicality can only be assessed with respect to the values observed in a population of control data.
For instance, the two SDs below the mean approach of Adcock and of Seghier (discussed at lines 368-370) also require (and rightfully so) collection of a control data set, from which the standard deviation can be derived.
Collecting a normative comparison group is a 'limitation' of all of the data driven approaches (i.e. those discussed above, as well as the Berl et al and the Mazoyer et al studies discussed next in the manuscript), and it is not really a 'limitation'. It would seem better characterised as an additional 'cost' (or some similar term) associated with data driven approaches, but this cost is undertaken because of the advantages that it confers on interpretation.

Additional comments

Overall I enjoyed reading this paper. I found it to be clear, well written, and a useful summary of the literature. It will be important to complement this paper with those proposed by the authors (lines 93-94) dealing with the importance of language task and baselining.

·

Basic reporting

no comment

Experimental design

no comment

Validity of the findings

no comment

Additional comments

In the present study, Bradshaw et al. investigated the different methods of quantifying and classifying laterality that have been used in fMRI studies since 2000. Aim of the study was to determine which procedures give the most robust and reliable measurement. The manuscript deals with an important topic and is well written. The referenced literature is relevant, the data analyses seem to be robust and sound. The manuscript is a good extension pf previous work from, e.g., Jansen et al. (2006) and Seghier et al. (2008) who previously gave an overview on different calculation methods for laterality indices. I just have a few minor comments:

p. 9: The authors describe that “the majority of studies used the standard LI ratio approach using the LI formula as previously outlined (55 within our search), but use of bootstrapping approaches started to be seen in 2010 and has gained in popularity since that time.” One reason for the increase in bootstrapping approaches might be that these approaches are nicely implemented in the SPM toolbox from Wilke and colleagues. The authors might want to describe which approaches are easily available. This might be one of the reasons why specific approaches gained popularity.

p. 17: The authors discuss the use of laterality indices assessing either the degree of lateralization or hemispheric dominance (i.e., right, left, bilateral). With regard to hemispheric dominance, they might want to mention that it is typically possible to determine left- and right-hemispheric dominance reproducibly, but not bilateral dominance (see e.g. Jansen et al, 2006).

p. 18: The flip method, as correctly stated by the authors, was developed to describe whether activity in one hemisphere is significantly stronger than activity in the other hemisphere. It can be used to determine significant left- or right-hemispheric dominance. It can however not be used to describe bilaterality, but only not-significantly lateralized activity. This should be stated in the manuscript. The authors further describe that “the resulting significant voxels in each hemisphere can then be used as input for the standard LI equation”. The authors are not wrong, of course. I nevertheless disagree with that statement. Typically, only one hemisphere has significant voxel, thus leading to LIs of +/- 1. It is, in my opinion, not easily possible to use the flip method to also describe meaninfully the degree of hemispheric lateralization.

As a last comment, I think the author should, at least shortly, discuss that the analysis of the lateralization of a cognitive function is not the endpoint of an analysis, since it does not use the full potential of the imaging data. In laterality research, in particular fMRI data might be further used to look at other important characteristics of lateralization, for instance intra- and interhemispheric connectivity (Stephan et al. 2003, 2005, 2007; Fräßle et al., 2016, 2017; Wende et al., 2017; Seghier et al., 2011).

---

## Round 0.2 · accepted · Accept

The concerns have been properly addressed.

·

Basic reporting

.

Experimental design

.

Validity of the findings

.

Additional comments

The authors responded to all criticism raised by the reviewers. I enjoyed reading the manuscript.